# Meta-Adapter: An Online Few-shot Learner for Vision-Language Model

**Cheng Cheng**[1]*   **Lin Song**[2]*   **Ruoyi Xue**[1]   **Hang Wang**[1]
**Hongbin Sun**[1]†   **Yixiao Ge**[2]   **Ying Shan**[2]
[1] Xi'an JiaoTong University   [2]Tencent AI Lab
cheng2016@stu.xjtu.edu.cn, ronnysong@tencent.com

## Abstract

The contrastive vision-language pre-training, known as CLIP, demonstrates remarkable potential in perceiving open-world visual concepts, enabling effective zero-shot image recognition. Nevertheless, few-shot learning methods based on CLIP typically require offline fine-tuning of the parameters on few-shot samples, resulting in longer inference time and the risk of over-fitting in certain domains. To tackle these challenges, we propose the Meta-Adapter, a lightweight residual-style adapter, to refine the CLIP features guided by the few-shot samples in an online manner. With a few training samples, our method can enable effective few-shot learning capabilities and generalize to unseen data or tasks without additional fine-tuning, achieving competitive performance and high efficiency. Without bells and whistles, our approach outperforms the state-of-the-art online few-shot learning method by an average of 3.6% on eight image classification datasets with higher inference speed. Furthermore, our model is simple and flexible, serving as a plug-and-play module directly applicable to downstream tasks. Without further fine-tuning, Meta-Adapter obtains notable performance improvements in open-vocabulary object detection and segmentation tasks.

## 1 Introduction

The contrastive vision-language pre-training [1–5], known as CLIP [6], has shown the impressive potential in modeling open-world visual concepts [7–17], which benefits multiple vision tasks including image recognition and open-vocabulary perception [7, 10]. It can be mainly attributed to the large-scale datasets [2] and the advanced pre-learning techniques [18]. By constructing prompts based on visual categories, CLIP shows effective zero-shot image classification capabilities and generalization abilities for unseen data [19]. Recently, few-shot learning based on CLIP has garnered increasing research attention. Motivated by the success of feature adapters [20] and prompt tuning [21] for natural language processing, a wide range of few-shot approaches for CLIP [22, 8, 9, 7] are proposed and studied.

Few-shot learning methods for CLIP can be categorized into offline [22, 8, 9] and online approaches [7], according to whether the fine-tuning is required for the few-shot samples of unseen categories. Offline methods extract the knowledge from few-shot samples via parameter optimization. Notable examples include CoOp [8] and CoCoOp [22], which replace the hand-crafted templates in CLIP with learnable continuous tokens by fine-tuning on few-shot samples. Additionally, CLIP-Adapter [9] introduces feature adapters into CLIP by learning task-specific knowledge from few-shot samples. Although the extra components yield promising few-shot learning capabilities, they also incur

---

[1]Equal contribution.
[2]Corresponding author.

37th Conference on Neural Information Processing Systems (NeurIPS 2023).

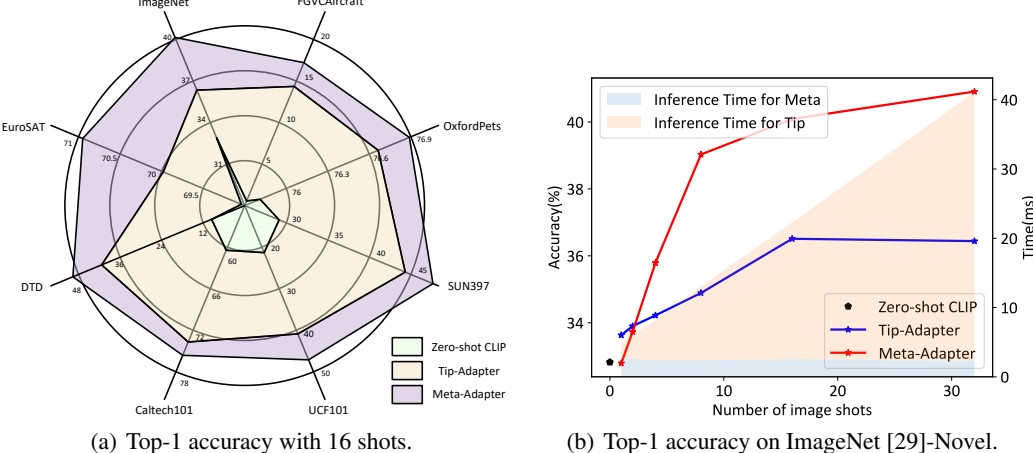

(a) Top-1 accuracy with 16 shots.

(b) Top-1 accuracy on ImageNet [29]-Novel.

Figure 1: Comparison of different few-shot learning techniques. The models are trained on the set of base classes and evaluated on the novel classes. The time is measured on a Tesla V100 GPU.

additional training costs and suffer from considerable over-fitting in a certain data distribution [22, 23]. To eliminate training expenses, an online method called Tip-Adapter [7] is proposed. This method proposes a hand-crafted modulation function that adjusts the ratio between category embeddings and few-shot visual embeddings. It obtains the knowledge from few-shot samples without fine-tuning and shows a notable improvement against the zero-shot manner. However, due to its sophisticated hyper-parameter search scheme, we find that Tip-Adapter still tends to over-fit in the distribution of seen data (details referring to Sec. 3.1), resulting in limited generalization capability. Different from previous methods, we attempt to explore a new perceptive: *learning an online few-shot learner for CLIP via meta-learning.*

To achieve it, we propose the Meta-Adapter that replaces the hand-crafted modulation function and searching scheme in Tip-Adapter with a lightweight residual-style network. The offline few-shot learning methods require additional fine-tuning for few-shot samples from unseen categories. In contrast, our approach employs the meta-testing mechanism [24–27], thereby the categories of training and testing data of our model can be different. By using a limited number of few-shot data, the Meta-Adapter can be trained to enable few-shot learning capability. Without additional fine-tuning, it can further generalize to other unseen data and extract knowledge from the few-shot samples in an online manner. To achieve high efficiency, the Meta-Adapter is constructed by a lightweight network based on the gated multi-head attention mechanism [28], which bridges the gap between few-shot image features and textual features for each category. This procedure can be considered as a learnable filter to refine the category embeddings guided by the few-shot images. Since the Meta-Adapter does not require additional fine-tuning, it only has a slight computational overhead over the zero-shot manner. Compared with the Tip-Adapter, it alleviates the over-fitting problem and demonstrates superior generalization across datasets. Furthermore, the Meta-Adapter is simple and can be applied as a plug-and-play module to various CLIP-based methods, making it a versatile solution for many open-vocabulary downstream tasks.

The extensive experiments demonstrate the effectiveness and efficiency of the Meta-Adapter on image classification, object detection, and segmentation. To verify the generalizability of Meta-Adapter, we conduct a series of ablation studies, including cross-category generalization within a certain dataset, cross-dataset generalization, and cross-task generalization which explores the potential of Meta-Adapter in downstream tasks. As shown in Figure 1(a), by training on the data of base classes, the Meta-Adapter achieves an average of 3.6% absolute gains over Tip-Adapter across the novel classes of eight image classification datasets under the 16-shot setting. With a larger number of image shots, our method achieves increased performance gain over the Tip-Adapter, which is illustrated in Figure 1(b). Besides, through directly evaluating the ImageNet [29] pre-trained model on other seven classification datasets, our method obtains an average of 4.9% improvements against Tip-Adapter. In addition, Meta-Adapter shows the potential to improve other tasks, such as open-vocabulary object detection, which leads to consistent improvements in both object detection and instance segmentation.

By integrating the ImageNet-pretrained Meta-Adapter with the open-vocabulary object detection framework, ViLD [11], our method achieves 1.0% absolute gains on the average precision of rare categories $AP_r$ without bells and whistles.

## 2 Related Work

### 2.1 Vision-Language Pretrained Models

Inspired by the success of pre-trained models in the field of CV and NLP, many works are proposed to pre-train large-scale models to process both vision and language modalities. A typical vision-language model consists of four key components, i.e., vision encoder, language encoder, fusion encoder, and loss function. Recently, following the success of the base models in both CV and NLP [30, 6, 31–36], the community of multi-modal learning can take advantage of these large-scale base models to better elevate the performance. VisualBERT [37], OSCAR [38], Uniter [39] utilize BERT [30] to preprocess the raw text and demonstrate impressive results in multimodal tasks, e.g., visual question answering (VQA). Besides, these methods require a well-designed fusion encoder to integrate the cross-modal interaction. Recently, CLIP [2], DeCLIP [40] and ALIGN [1] demonstrate that vision-language contrastive learning is capable of generating transferable features to downstream tasks and the multimodal interaction can be well interpreted by simply calculating the dot product between vision and language embeddings. Without additional self-attention or cross-attention modules, the multimodal embeddings can be pre-computed and stored, which is more efficient and can be easily adapted to other tasks.

### 2.2 Vision-Language Model Adaption

Many recent works focus on exploring effective and efficient approaches for adapting vision-language models to downstream tasks [8, 22, 7, 9, 41, 42, 14, 13, 15, 43, 12], which are prompt-tuning methods, e.g., Context-Optimization (CoOp) [8], and feature adapters methods, e.g., Tip-Adapter [7, 14, 13]. Inspired by the success of prompt learning [21, 44], CoOp proposes to replace the hand-crafted templates [2] with continuous tokens that can be optimized in a traditional fine-tuning fashion. Besides, to mitigate the woeful over-fitting issue, CoCoOp further introduces integrate image-specific tokens learned by a shallow MLP. Compared with the manually designed prompt templates, CoOp and CoCoOp achieve impressive performance on few-shot image classification. Different from these prompt tuning methods, CLIP-Adapter and Tip-Adapter conduct residual feature blending to integrate few-shot knowledge with CLIP's zero-shot knowledge. They keep the whole CLIP's parameters frozen and fine-tune an acceptable small number of additional weights and show impressive results in few-shot image classification. Besides, by initializing the linear weights with few-shot knowledge (i.e., cache model in this context), Tip-Adapter can further pose a training-free fashion with preferable performance. Nevertheless, these methods suffer from over-fitting, especially when the domain gap between the source and target datasets is large.

### 2.3 Meta-Learning

A simple interpretation of meta-learning [24–27] is "learning-to-learn" [45] which corresponds to improving the generalization by searching for the algorithm (inductive bias) that is best suited for a given task family. On the contrary, traditional machine learning algorithms [46, 47] are expected to be improved as more data from a certain single task. Usually, meta-learning is conducted on learning instances sampled from a task family, which is expected to simulate a base learning algorithm that performs well on new tasks sampled from this family. Besides, as mentioned in [24], all training instances can be sampled from a single task in a special case. In the context of adapting the vision-language model to downstream tasks, meta-learning can be viewed as learning the general fine-tuning algorithms which bring consistent gains over different tasks or datasets. Current methods [22, 8, 7, 9] focus mainly on improving the performance of certain tasks or datasets. To the best of our knowledge, this paper is the first one that studies the potential of meta-learning in the field of vision-language model adaption.

Table 1: Comparison of cross-dataset generalization based on ImageNet [29] pre-training. The Tip-Adapter and Meta-Adapter are tuned on ImageNet and frozen for other datasets. $\Delta$ reflects the generalization ability across datasets.

| Method | FGVC | OxfordPets | SUN397 | UCF101 | Caltech101 | DTD | EuroSAT | Avg. | $\Delta$ |
|---|---|---|---|---|---|---|---|---|---|
| Zero-shot CLIP | 0.42 | 56.25 | 28.96 | 21.05 | 60.62 | 10.00 | 4.17 | 25.92 | - |
| Tip-Adapter* | 13.96 | 68.75 | 45.16 | 40.09 | 68.33 | 42.92 | 56.25 | 47.92 | - |
| Tip-Adapter | 13.96 | 67.19 | 43.80 | 39.47 | 67.08 | 40.00 | 56.25 | 46.82 | - |
| Meta-Adapter* | 19.58 | 72.66 | 51.25 | 52.28 | 71.46 | 49.17 | 64.58 | 54.43 | +6.51 |
| Meta-Adapter | **15.21** | **72.66** | **48.54** | **47.54** | **67.92** | **48.33** | **62.50** | **51.81** | **+4.99** |

\* indicates searching hyper-parameter or training for each evaluation dataset individually.

## 3 Method

In this section, we introduce the proposed Meta-Adapter. In Section 3.1, we first revisit CLIP and Tip-Adapter. In Section 3.2, we elaborate on the implementation of the proposed Meta-Adapter. In Section 3.3, we discuss the difference with other related works.

### 3.1 Revisiting CLIP and Tip-Adapter

As a vision-language pre-training model, CLIP [2] has shown impressive zero-shot learning potential in modeling open-world visual representation [11, 10] by exploiting contrastive learning with large-scale noisy image-text pairs. To achieve zero-shot image classification, CLIP computes classification scores by measuring the cosine distance between the image features and per-class textual features. Specifically, given an image $y$, let $f \in \mathbb{R}^{D \times 1}$ be the feature of the query image and $\{w_i\}_{i=1}^{N}, w_i \in \mathbb{R}^{D \times 1}$ be a set of category embeddings generated by the text encoder. The $D$ indicates the dimension of embedding space and $N$ denotes the number of total categories. The textual feature $w_i$ for each class is derived from hand-crafted templates that one of typical form is "a photo of [CLASS]". The class token is then replaced by a specific category name, such as "Alp" or "Lemon", as shown in Figure 2. The predicted logits of the given image $y$ belonging to the $i$-th class can be formulated as:

$$\text{logits}(y_c = i) = \frac{w_i^\top f}{\|w_i\| \|f\|}, \tag{1}$$

Tip-Adapter [7] further proposes an online method to learn knowledge from few-shot samples. This method employs a straightforward modulation function with a stochastic hyper-parameter search strategy, achieving impressive few-shot performance for a certain domain. Concretely, two hyper-parameters, i.e., $\alpha$, and $\beta$, are introduced to adjust the ratio between visual and textual features for different datasets. Given a set of support images $\mathbf{x} = \{\mathbf{x}_i\}_{i=1}^{N}$ in $N$ ways and $K$ shots, the predicted logits of Tip-Adapter can be formulated as :

$$\text{logits}(y_c = i | \mathbf{x}, \alpha, \beta) = \frac{w_i^\top f}{\|w_i\| \|f\|} + \alpha \cdot \exp(-\beta(1 - \frac{\mathbf{F}_j^\top f}{\|\mathbf{F}_j\| \|f\|})) \mathbf{L}_j, \tag{2}$$

where $\mathbf{F}_i \in \mathbb{R}^{D \times K}$ is support embeddings of few-shot samples, and $\mathbf{L}_i \in \mathbb{R}^{N \times K}$ is the corresponding one-hot labels of $i$-th class. Tip-Adapter obtains remarkable few-shot performance without additional training for few-shot samples. However, Tip-Adapter relies heavily on the hyper-parameter search strategy on the target dataset, rendering it susceptible to over-fitting within a certain data distribution and limiting its out-of-distribution generalization capabilities. As presented in Table 1, we fix the hyper-parameters searched on ImageNet [29] and directly evaluate the performance of Tip-Adapter on seven other datasets. It demonstrates a poor generalization performance of the Tip-Adapter across different distributions. Compared with the searching scheme for each dataset individually, the Tip-Adapter exhibits a notable decline in performance.

### 3.2 Meta-Adapter

To address the issue of poor generalization, we propose a learnable Meta-Adapter to replace the handcraft modulation function and searching strategy in Tip-Adapter. Different from the Tip-Adapter,

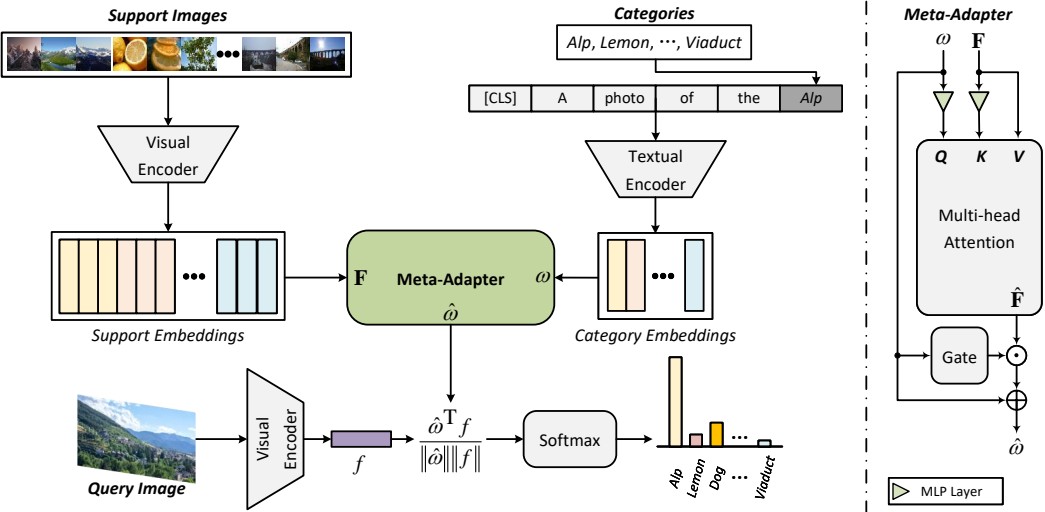

Figure 2: Diagram of the proposed Meta-Adapter, which employs a learnable network to refine the category embeddings guided by few-shot images.

we model the few-shot learning in a visual-language scheme as a learnable filter on textual features under the guidance of few-shot image samples, to obtain more discriminative category embeddings. Motivated by the non-local filters [48–50] in computer vision, we propose a Meta-Adapter based on the gated multi-head attention mechanism.

As shown in Figure 2, through CLIP encoders, we first extract the support embeddings of input few-shot images and category embedding. The Meta-Adapter then extracts and transfers the few-shot knowledge from visual features into textual features to obtain refined category embeddings. Specifically, we apply the original category embeddings as query and the support embeddings as both key and value into a multi-head attention block. Unlike the standard transformer encoder [28], our approach only introduces Multilayer Perceptron (MLP) layers for query and key. This strategy is crucial since no feature transformations are performed on values, the zero-shot capability is generally not changed after training. The predicted logits with the proposed Meta-Adapter can be formulated as:

$$\text{logits}(y_c = i|\mathbf{x}) = \frac{\hat{w}_i^\top f}{\|\hat{w}_i\| \|f\|}, \text{ where } \hat{w} = \text{MetaAdapter}(w, \mathbf{F}). \tag{3}$$

The $\hat{w}$ is the refined category embeddings. In the Meta-Adapter, as shown in the right-hand of Figure 2, the proposed method adaptively aggregate the support embeddings according to the affinity between categories and few-shot images. The aforementioned procedure can be implemented by a cross-attention mechanism:

$$\hat{\mathbf{F}} = \mathbf{F}^\top \sigma((\mathbf{F}W_1^\top)(wW_2^\top)^\top / \sqrt{D}) \tag{4}$$

where $W_1$ and $W_2$ indicate the weights of MLP layers. The $\sigma$ denotes the Softmax function, and $\hat{\mathbf{F}}$ represents the aggregated support features. Intuitively, similar to the non-local filters, Meta-Adapter could disregard some outlier samples while paying more attention to the samples that are more related to the category description [42], resulting in robust feature representations.

Besides, the importance of textual and visual features for few-shot learning varies across different data distributions [9]. Therefore, we propose a learnable gating block $g(\cdot)$, generating a modulation scalar, to adaptively control the ratio between category embeddings and aggregated support embeddings. Accordingly, the refined category embedding can be obtained by:

$$\hat{w} = w + g(w) \odot \hat{\mathbf{F}}, \tag{5}$$

where $\odot$ denotes Hadamard product. Through training on the few-shot samples, the gating block could adjust the ratio according to the category descriptions. It enables the proposed method to effectively integrated few-shot knowledge with zero-shot knowledge.

Table 2: Quantitative results of in-domain generalization setting on UCF101, Caltech101, DTD, and FGVCAircraft datasets between Meta-Adapter and other methods.

| Model | UCF101 | | Caltech101 | | DTD | | FGVCAircraft | |
|---|---|---|---|---|---|---|---|---|
| | Base | Novel | Base | Novel | Base | Novel | Base | Novel |
| Zero-shot CLIP | 79.42 | 21.05 | 93.39 | 60.62 | 59.38 | 10.00 | 23.84 | 0.42 |
| Tip-Adapter | 85.17 | 40.09 | 95.09 | 68.33 | 68.36 | 42.92 | 30.27 | 13.96 |
| Meta-Adapter | 82.44 | **52.28** | 93.39 | **71.46** | 64.26 | **49.17** | 27.32 | **19.58** |

## 3.3 Comparison with Counterparts

Compared to the offline methods, e.g., CLIP-Adapter [9] and CoOp [8], our Meta-Adapter does not require additional fine-tuning for target samples, significantly reducing the computational costs during inference. In addition, compared with the online methods, e.g., Tip-Adapter [7], the proposed technique replaces the handcrafted hyper-parameter search process with a learnable network on support samples. As shown in Table 1, the Meta-Adapter can better alleviates the over-fitting problem and demonstrates the generalization across datasets without further fine-tuning. Moreover, as the Meta-Adapter refines the textual embedding features directly without altering their dimensions, it can naturally be applied to a variety of downstream tasks based on CLIP.

# 4 Experiments

It should be noted that the distributions of training and testing sets can be identical or dissimilar and it is crucial that Meta-Adapter can well perform in both scenarios. Besides, the potential of Meta-Adapter in downstream tasks is also of great importance. We refer to these three situations as "*cross-category generalization*", "*cross-dataset generalization*", and "*cross-task generalization*", respectively. Specifically, for "*cross-category generalization*", we split the full categories of each dataset into base and novel sets according to the per-category accuracy predicted by Zero-shot CLIP, that is, the base set contains easy samples and the novel set contains hard samples. This dataset split strategy simulates a rather difficult situation to verify whether Meta-Adapter is able to learn the dataset-irrelevant approach, especially for hard samples. We provide details of dataset splits in the supplementary material. Before diving into experimental analysis, we first give the details of the experimental setup.

**Datasets** For cross-category generalization experiments, we use 8 representative image classification datasets: ImageNet [29], FGVCAircraft [51], OxfordPets [52], SUN397 [53], UCF101 [54], Caltech101 [55], DTD [56], and EuroSAT [57], which cover a diverse set of classification tasks. As for the cross-dataset generalization experiment, ImageNet is further utilized as the source dataset and its three variants are treated as target datasets, i.e., ImageNet-A [58], ImageNet-R [59], and ImageNet-Sketch [60]. Moreover, to explore the potential of Meta-Adapter on open-vocabulary detection, we conduct experiments on LVIS [61].

**Baselines** We compare Meta-Adapter with two training-free methods: Zero-shot CLIP [2] and Tip-Adapter [7].

**Training Details** As for the CLIP backbone, we choose ResNet50 [46] as the visual encoder in most experiments and a transformer [6] as the textual encoder. We adopt the prompt ensemble strategy [2, 7], which inputs 7 templates into the CLIP textual encoder and then averages them as the final prompt embeddings. We optimize the Meta-Adapter on the base set with a batch size of 64 and use AdamW optimizer [62] with a learning rate of 0.0001 and a cosine scheduler for 5 epochs.

## 4.1 Cross-Category Generalization

Based on empirical studies, it can be observed that Tip-Adapter often requires large hyper-parameters ($\alpha$ and $\beta$) when applied to specific datasets. These hyper-parameters are used to smooth the classification distribution and give significant weight to few-shot knowledge. This phenomenon indicates that Tip-Adapter heavily relies on few-shot knowledge, even for relatively general datasets like ImageNet. As a result, over-fitting issues arise, which hinder its generalization ability. Table 6 demonstrates that

Table 3: Quantitative results on ImageNet of different models utilized various vision backbones.

| Model | RN50 | RN101 | ViT-B/32 | ViT-B/16 | RN50×16 | RN50×64 |
|---|---|---|---|---|---|---|
| Zero-shot CLIP | 32.82 | 39.22 | 40.10 | 45.77 | 50.10 | 54.67 |
| Tip-Adapter | 36.51 | 42.42 | 43.71 | 49.84 | 53.08 | 57.99 |
| Meta-Adapter | **40.19** | **47.01** | **46.91** | **52.60** | **55.51** | **60.41** |

Tip-Adapter achieves slightly higher classification accuracy than Meta-Adapter on the training set for datasets like UCF101 and Caltech101. However, when it comes to novel samples, Tip-Adapter falls behind Meta-Adapter with significant gaps, such as 40.26% versus 47.72% on UCF101. This suggests that Tip-Adapter becomes excessively tailored to a specific localized distribution due to its excessive hyper-parameter search strategy.

On the contrary, thanks to the ingenious and generic ensemble approach, Meta-Adapter enjoys comparable performances on the base set and superior performances on the novel set. As shown in Figure 1(b), Meta-Adapter shows superior performances over other methods on the ImageNet dataset. Compared to Zero-shot CLIP, Meta-Adapter consistently surpasses it on all different few-shot settings. In comparison to Tip-Adapter, the classification accuracy of both methods witnesses relatively steady improvements. When shots are less than 4, Tip-Adapter slightly outperforms Meta-Adapter. The reason is two-fold. First, Tip-Adapter directly calculates classification logits given both few-shot features and their corresponding one-hot labels, which can be viewed as a shortcut solution compared to the generic ensemble approach of Meta-Adapter. Second, Tip-Adapter exploits the potential of few-shot knowledge via a hyper-parameter searching strategy which intends to find out the highest accuracy on certain datasets. Nevertheless, as shots get larger, Meta-Adapter outperforms Tip-Adapter by a clear margin and classification accuracy goes up consistently while Tip-Adapter witnesses performance drops when shots are 32, indicating a possible performance limitation for Tip-Adapter. And we present the quantitative comparison on the other 7 datasets under 16 shots setting in Figure 1(a). It can be observed that Meta-Adapter significantly boosts the classification accuracy over Zero-shot CLIP and surpasses Tip-Adapter with gains of up to +7.46%. Due to space limitations, we present the comparison of Meta-Adapter and other methods on the remaining seven datasets under different few-shot settings in the supplementary material.

To further verify the effectiveness of Meta-Adapter, we apply different visual encoders for all methods and conduct experiments on ImageNet. The quantitative comparison is shown in Table 3. Undoubtedly, Meta-Adapter maintains its leading position over Tip-Adapter regardless of the choice of visual encoders. As a more advanced backbone, such as ViT-B/16 [2], is utilized, the classification accuracy of Meta-Adapter further increases. This suggests that the learning potential of Meta-Adapter can be enhanced by employing a more powerful vision-language model. In summary, compared to previous training-free methods, Meta-Adapter not only effectively mitigates the issue of over-fitting but also retains superior generalization ability. As a result, it achieves state-of-the-art classification accuracy on the novel set.

### 4.2 Cross-Dataset Generalization

It is of great importance that a learned classifier maintains comparable performance when handling different datasets with diverse distributions. It is more challenging because the appearances and shapes can be totally dissimilar across datasets (e.g., from object recognition of ImageNet to textual classification of DTD). Besides, we are interested to know whether Meta-Adapter can learn dataset-irrelevant discriminative ensemble approach. To this end, we transfer the optimal hyper-parameters of Tip-Adapter searched on the source dataset and evaluate the performance on target datasets. And Meta-Adapter is first trained on the source dataset, i.e., base set, and evaluated on the target dataset, i.e., novel set, with all learnable parameters frozen. Both experiments are conducted under the 16-shot setting. We report the relative accuracy improvements when transferring from ImageNet (with all categories utilized) to the other 7 datasets, as shown in Figure 3(a). Tip-Adapter base results are set as the baseline in Figure 3(a), thus the corresponding relative accuracy improvements are always 1.0. Considering that ImageNet contains a variety of classes, e.g., different animal breeds and different kinds of vehicles, it is not surprising that both models retain comparable accuracy compared with their counterparts. Nevertheless, Tip-Adapter poses rather clear performance drops compared with Meta-Adapter. Moreover, transferring from ImageNet to SUN397, UCF101, and EuroSAT

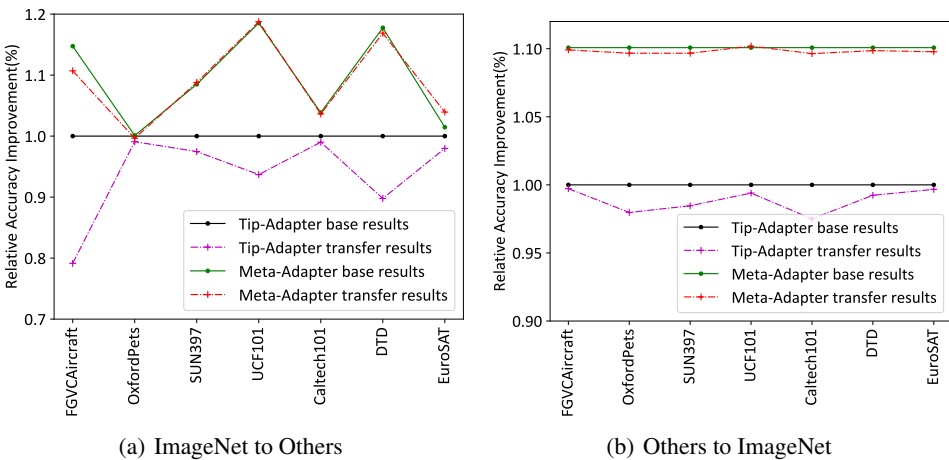

(a) ImageNet to Others        (b) Others to ImageNet

Figure 3: Relative accuracy improvements of Tip-Adapter and Meta-Adapter in cross-dataset generalization experiments.

Table 4: Quantitative results of domain generalization experiments between Tip-Adapter and Meta-Adapter. The data in parentheses records the changes brought by comparing with Zero-shot CLIP.

| CLIP Backbone | Model | Target Datasets | | |
| --- | --- | --- | --- | --- |
| | | ImageNet-A | ImageNet-R | ImageNet-Sketch |
| RN50 | Zero-shot CLIP | 23.88 | 60.54 | 35.45 |
| | Tip-Adapter | 23.25(-0.63) | 58.73(-1.81) | 34.77(-0.68) |
| | Meta-Adapter | 23.71(-0.17) | 59.96(-0.58) | 35.54(+0.09) |
| ViT-B/16 | Zero-shot CLIP | 50.65 | 77.82 | 48.42 |
| | Tip-Adapter | 49.89(-0.76) | 76.94(-0.88) | 48.13(-0.29) |
| | Meta-Adapter | 51.12(+0.47) | 77.54(-0.28) | 48.76(+0.34) |

results in surpassing their baselines, i.e., directly trained on the target datasets, which suggests that the learning potential of Meta-Adapter can benefit from a generalized dataset. Besides, we report the relative accuracy improvements of Tip-Adapter and Meta-Adapter by transferring from 7 small classification datasets (with all available categories utilized) to ImageNet, as shown in Figure 3(b). Surprisingly, Meta-Adapter maintains comparable results while Tip-Adapter poses clear performance drops, particularly on OxfordPets and Caltech101. Thus, it can be concluded that Meta-Adapter retains better transferability against diverse distributions and domain shifts.

Furthermore, we conduct domain generalization experiments as in [22]. Humans have a natural ability to generalize to out-of-distribution data, which raises the question of whether Meta-Adapter possesses the same advantage. To this end, we transfer Tip-Adapter's optimal hyper-parameters ($\alpha$ and $\beta$) and Meta-Adapter's weights searched and optimized on ImageNet to its three variants, i.e., ImageNet-A [58], ImageNet-R [59], and ImageNet-Sketch [60]. Besides, we report the performance of Zero-shot CLIP on these three datasets as the baseline. The quantitative results are presented in Table 4. It is clear that directly transferring Tip-Adapter's optimal hyper-parameters searched on ImageNet to its variants leads to performance drops, even worse than Zero-shot CLIP. This phenomenon accords with the previous statement that Tip-Adapter is sensitive to hyper-parameter settings, or to say Tip-Adapter suffers from severe over-fitting issues. On the contrary, Meta-Adapter can better adapt to domain shifts, demonstrating comparable performances with Zero-shot CLIP.

### 4.3 Cross-Task Generalization

It is a major concern whether few-shot learning methods can benefit downstream tasks. To this end, we integrate Meta-Adapter and Tip-Adapter with open-vocabulary object detection method ViLD [11]. ViLD utilizes CLIP to explore the open-vocabulary potential, it replaces the typical classifier in

Table 5: Comparison of different few-shot methods on LVIS [61] val set.

| Methods | Object Detection | | | Instance Segmentation | | |
|---|---|---|---|---|---|---|
| | $AP_r$ | $AP_c$ | $AP_f$ | $AP_r$ | $AP_c$ | $AP_f$ |
| ViLD | 18.1 | 27.3 | 32.0 | 17.4 | 25.5 | 28.5 |
| ViLD + Tip-Adapter | 11.3 | 23.3 | 26.7 | 10.8 | 21.8 | 24.2 |
| ViLD + Meta-Adapter | **19.1** | 27.3 | 31.7 | **18.3** | 25.5 | 28.3 |

object detection framework [63, 47, 64, 65, 17, 66–69] with textual features generated by CLIP's text encoder and aligns the textual features and ROI features by knowledge distillation.

We first generate pre-processed region features of LVIS [61] given its annotations as the few-shot samples. Following ViLD, object categories in the LVIS dataset are split into "frequent", "common", and "rare" according to their frequency in the training set. Among them, 866 frequent and common categories of LVIS are taken as the base set, and 337 rare categories are held out as the novel set. For Tip-Adapter, the prediction logits of ViLD is modified by adding an additional few-shot term.

$$\text{logits}(\hat{r}) = f(\hat{r})w^T + \alpha \cdot \varphi(f(\hat{r})\mathbf{F}^\top)\mathbf{L}, \text{ where } \varphi(x) = \exp(-\beta(1 - x)) \tag{6}$$

where $f(\hat{r})$ represents the region features of proposal $\hat{r}$; $\varphi(\cdot)$ is the modulation function introduced in Tip-Adapter; $\mathbf{F}$ and $\mathbf{L}$ denotes few-shot region features and their corresponding one-hot labels. It should be noted that Equation 6 is applied to both ViLD-text and ViLD-image as introduced in ViLD. As for the hyper-parameter settings ($\alpha$ and $\beta$) of Tip-Adapter, we observe that setting them to be the ones that are searched on ImageNet will cause ViLD collapses. Thus, we set $\alpha$ to 0.05 and $\beta$ to 1, which encourages ViLD to rely more on CLIP knowledge to avoid collapse. Besides, we utilize the ImageNet pre-trained Meta-Adapter to integrate the pre-processed LVIS few-shot knowledge into the original textual features. And for ViLD, we utilize the re-implement version as suggested in DetPro [10] which replaces the pre-trained ResNet-50 [46] with self-supervised pre-trained SoCo [70]. We report the average precision in Table 5. From Table 5, it is clear that Meta-Adapter can boost the detection ability on rare categories by a clear margin while Tip-Adapter corrupts the detection performances due to its poor transferability and its necessity to modify the original prediction score of ViLD. Besides, considering that part of the LVIS annotations contains rather small bounding boxes (usually around $10 \times 10$), the quality of the LVIS few-shot dataset may not be as good as the image classification counterparts. As mentioned before, Tip-Adapter naturally and heavily relies on few-shot knowledge, which may be one of the main causes of poor performance in open-vocabulary object detection. On the contrary, Meta-Adapter learns the generic ensemble approach and can be easily integrated into open-vocabulary object detection methods without changing their formulation of prediction scores.

Table 6: Comparison of Zero-Shot CLIP, Tip-Adapter, and Meta-Adapter on UCF101 and Caltech101 datasets in in-domain generalization setting. H: Harmonic mean (to highlight the generalization trade-off [22]).

| Dataset | ImageNet | | | UCF101 | | | Caltech101 | | |
|---|---|---|---|---|---|---|---|---|---|
| Model | Base | Novel | H | Base | Novel | H | Base | Novel | H |
| Zero-shot CLIP | 71.9 | 32.8 | 45.0 | 79.4 | 21.1 | 33.4 | 95.4 | 60.6 | 74.1 |
| CLIP-Adapter | 76.3 | 15.1 | 25.3 | 89.4 | 5.4 | 10.2 | 97.3 | 39.3 | 54.0 |
| CoOp | 75.3 | 2.7 | 5.2 | 89.3 | 1.0 | 2.0 | 97.2 | 31.3 | 47.4 |
| CoCoOp | 75.5 | 33.9 | 46.8 | 86.5 | 9.1 | 16.5 | 96.8 | 60.9 | 74.8 |
| Meta-Adapter | 76.3 | 40.8 | 53.2 | 82.4 | 47.7 | 0.4 | 94.9 | 76.1 | 84.4 |

## 4.4 Comparison with Offline Methods

As shown in Table 6, we provide the ablation studies between our meta-adapter and other offline methods [22, 8, 9]. For a fair comparison, similar to CoCoOp, the experiments adopt a base-to-novel generalization setting. The results demonstrate that our method has significant advantages in generalization on novel classes, e.g. improving over CoCoOp by 6.9% on ImageNet. Additionally,

as in previous works, we introduce Harmonic Mean to measure overall performance on base and novel classes. It can be observed that our approach also shows clear superiority in terms of overall performance. More importantly, finetuning is not needed for our method when applying to new datasets or tasks.

## 4.5 More ablation studies of the Meta-Adapter

**A2:** Thanks for this insightful suggestion. Accordingly, we further conduct more ablation studies to demonstrate the advantages of our design. As shown in Table 7, the results demonstrate that multi-head attention contributes most significantly to improving accuracy. The proposed learnable gating block can further enhance the performance while introducing value projection leads to decreased generalization capability. Besides, as shown in Table 8, we increase the model scale of the meta-adapter by widening the projection layers (Wider) or cascading multiple modules (Deeper). The results show that increasing the number of modules can improve the parameter size and accuracy slightly, but brings a significant efficiency decrease.

Table 7: Ablation study on different components. The 'VP', 'MHA', and 'LGB' indicate the value projection layer, multi-head attention block, and learnable gating block, respectively.

| Method | ImageNet | SUN397 | UCF101 | DTD |
|---|---|---|---|---|
| Meta-Adapter w/ VP | 25.6 | 18.2 | 5.9 | 5.0 |
| Meta-Adapter w/o MHA | 32.9 | 28.9 | 21.4 | 10.0 |
| Meta-Adapter w/o LGB | 39.6 | 51.3 | 49.8 | 53.8 |
| Meta-Adapter | **40.2** | **52.7** | **51.4** | **54.6** |

Table 8: Comparison of different model scales on several datasets.

| Method | ImageNet | SUN397 | UCF101 | DTD | #Param | Latency |
|---|---|---|---|---|---|---|
| Meta-Adapter | 40.8 | 52.6 | 51.0 | 55.8 | 2.1M | 3ms |
| + Wider ($\times 2$) | 40.1 | 51.4 | 50.0 | 55.0 | 4.2M | 5ms |
| + Wider ($\times 4$) | 40.5 | 51.3 | 50.4 | 55.0 | 8.4M | 9ms |
| + Deeper ($\times 2$) | 40.4 | 52.9 | 52.2 | 56.7 | 4.2M | 6ms |
| + Deeper ($\times 4$) | 38.9 | 52.7 | 51.4 | 56.3 | 8.4M | 11ms |

## 5 Limitations and Conclusion

**Limitations** Based on the findings presented in Table 6, it can be observed that the learning potential of Meta-Adapter may encounter limitations when the classification accuracy of Zero-shot CLIP is high, as seen in the cases of UCF101 and Caltech101. The underlying reason behind this could be the combination of image-image similarity scores with text-image similarity scores, which might impede the potential of few-shot learning. In other words, in such scenarios, Meta-Adapter appears to favor zero-shot knowledge over few-shot knowledge, resulting in an imbalance between the two. Furthermore, empirical studies indicate that Meta-Adapter struggles to be applied to open-vocabulary semantic segmentation tasks where obtaining high-quality few-shot datasets is challenging. It is possible that incorporating external data could help alleviate this issue. These aforementioned challenges are left for future research and investigation.

**Conclusion** This paper demonstrates the significant potential of Meta-Adapter, a new few-shot learning method for CLIP, which is designed to overcome the limitations of previous methods in terms of poor generalization ability and low efficiency. The Meta-Adapter, employing a meta-testing mechanism and a lightweight residual-style network, extracts knowledge from few-shot samples without the need for additional fine-tuning, thus alleviating the over-fitting issue while maintaining high efficiency. Our findings highlight the impressive performance of the Meta-Adapter in various tasks, including image classification, object detection, and segmentation, indicating its superior generalization capabilities across datasets and tasks. Future work could focus on further refining the Meta-Adapter and exploring its potential applications in other vision tasks, advancing the capabilities of few-shot learning techniques in visual concept modeling.

# 6 Acknowledgments

This work was supported by the National Natural Science Foundation of China (No. 62204200).

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
