# 1 Details for Dataset Partitioning

Here we provide the dataset partitioning results for ImageNet [1], FGVCAircraft [2], OxfordPets [3], SUN397 [4], UCF101 [5], Caltech101 [6], DTD [7], and EuroSAT [8].

**ImageNet**

**Base categories indexes**: [986, 985, 668, 430, 14, 974, 685, 607, 537, 466, 90, 24, 993, 984, 933, 927, 800, 781, 679, 645, 573, 565, 510, 476, 340, 339, 333, 283, 95, 89, 983, 916, 820, 701, 614, 554, 458, 444, 400, 396, 323, 322, 145, 143, 69, 13, 0, 996, 959, 895, 890, 874, 802, 779, 425, 404, 399, 388, 284, 275, 139, 137, 98, 87, 989, 982, 964, 955, 926, 924, 922, 863, 805, 795, 755, 746, 640, 555, 535, 533, 500, 475, 351, 325, 293, 289, 255, 148, 144, 135, 15, 9, 992, 903, 873, 867, 832, 803, 739, 688, 671, 628, 625, 580, 574, 560, 547, 496, 332, 330, 321, 320, 292, 195, 149, 138, 76, 19, 11, 10, 980, 957, 937, 936, 917, 900, 878, 763, 687, 576, 564, 532, 471, 410, 383, 382, 346, 336, 294, 286, 268, 181, 131, 130, 118, 88, 51, 995, 965, 963, 946, 825, 766, 752, 719, 661, 611, 586, 546, 450, 449, 424, 407, 391, 352, 350, 324, 316, 309, 300, 291, 146, 91, 84, 82, 80, 48, 25, 18, 12, 8, 991, 953, 886, 822, 780, 736, 732, 682, 627, 557, 528, 524, 498, 486, 477, 474, 437, 403, 387, 367, 365, 363, 347, 344, 317, 306, 301, 259, 251, 147, 75, 16, 1, 994, 956, 938, 918, 915, 884, 775, 734, 703, 690, 672, 563, 548, 525, 511, 454, 395, 376, 354, 338, 305, 299, 205, 178, 152, 33, 981, 962, 958, 948, 945, 944, 934, 850, 847, 791, 783, 727, 605, 603, 568, 562, 520, 518, 467, 401, 393, 386, 295, 258, 111, 28, 22, 4, 952, 760, 743, 695, 694, 642, 610, 597, 594, 551, 540, 531, 483, 465, 342, 308, 296, 260, 223, 210, 150, 140, 127, 116, 105, 104, 102, 96, 70, 30, 950, 935, 932, 921, 919, 892, 880, 829, 768, 761, 713, 712, 711, 654, 639, 621, 595, 592, 561, 448, 440, 439, 420, 406, 398, 335, 307, 279, 254, 245, 213, 208, 132, 128, 108, 100, 31, 17, 997, 931, 888, 881, 866, 860, 853, 849, 833, 797, 741, 723, 652, 649, 637, 515, 433, 426, 402, 397, 349, 253, 252, 243, 235, 222, 180, 174, 171, 136, 133, 109, 107, 39, 29, 20, 951, 897, 865, 796, 759, 509, 443, 384, 355, 288, 276, 274, 247, 217, 194, 183, 141, 123, 92, 77, 68, 45, 37, 21, 913, 912, 858, 812, 786, 758, 756, 751, 714, 709, 697, 630, 575, 522, 491, 487, 480, 431, 421, 364, 357, 328, 261, 214, 153, 117, 74, 971, 967, 940, 907, 882, 872, 871, 862, 844, 843, 839, 827, 789, 726, 720, 646, 613, 570, 517, 495, 453, 392, 337, 329, 297, 287, 270, 249, 230, 203, 182, 161, 156, 142, 106, 81, 50, 973, 852, 835, 788, 717, 707, 704, 698, 643, 635, 629, 617, 577, 552, 543, 539, 468, 428, 422, 343, 327, 298, 234, 216, 190, 71, 61, 53, 34, 966, 939, 846, 831, 747, 686, 650, 620, 553, 526, 514, 436, 429, 366, 318, 273, 241, 209, 169, 115, 113, 72, 954, 941, 929, 901, 889, 883, 879, 877, 823, 798, 777, 770, 757, 669, 662, 660, 657, 647, 588, 571, 521, 470, 452, 442, 358, 334, 319, 239, 228, 207, 173, 159, 129, 125, 122, 86, 38, 869, 864, 793, 776, 769, 684, 666, 655, 634, 632, 615, 612, 608, 569, 559, 508, 484, 432, 378, 375, 370, 362, 348, 313, 302, 263, 262, 256, 237, 199, 176, 168, 120, 57, 56, 49, 41, 7, 2, 969, 949, 819, 806, 706, 674, 626, 616, 589, 513, 481, 462, 441, 427, 415, 379, 373, 361, 360, 244, 191, 93, 42, 5, 3, 990, 979, 977, 925, 894, 716, 675, 624, 606, 585, 538, 435, 423, 408, 405, 385, 304, 179, 175, 165, 164, 94, 85, 65, 961, 920, 857, 830, 826, 774, 738, 724, 692, 581, 534, 490, 485, 478, 472, 369, 311, 290, 232, 221, 218, 212, 162, 151, 97, 43, 35, 32, 26, 23, 987, 978, 943, 887, 861, 851, 821, 815, 764, 762, 740, 729, 699, 665, 636, 622, 601, 593, 591, 530, 507, 419, 394, 368, 233, 206, 198, 154, 112, 972, 960, 928, 909, 898]

**Novel categories indexes**: [891, 875, 854, 836, 801, 773, 631, 602, 584, 558, 541, 529, 489, 460, 451, 341, 303, 277, 271, 236, 202, 185, 184, 160, 126, 83, 64, 63, 46, 942, 910, 904, 893, 817, 808, 794, 785, 656, 651, 599, 598, 583, 582, 579, 572, 544, 497, 492, 417, 414, 380, 331, 281, 224, 196, 188, 103, 99, 44, 845, 834, 814, 811, 809, 807, 790, 737, 689, 678, 641, 578, 566, 549, 527, 506, 479, 457, 456, 413, 377, 372, 315, 310, 225, 197, 167, 124, 114, 79, 930, 856, 855, 841, 721, 590, 542, 459, 447, 390, 371, 272, 265, 220, 192, 170, 62, 27, 975, 914, 765, 735, 710, 708, 683, 503, 502, 501, 463, 455, 314, 264, 200, 121, 119, 52, 908, 896, 870, 799, 673, 663, 596, 494, 411, 246, 229, 211, 40, 36, 6, 911, 772, 754, 753, 728, 623, 523, 512, 280, 267, 227, 219, 177, 172, 166, 906, 859, 840, 804, 792, 748, 696, 691, 545, 504, 464, 434, 242, 238, 186, 110, 999, 976, 923, 745, 715, 567, 482, 473, 285, 266, 204, 187, 78, 988, 970, 968, 838, 667, 659, 644, 619, 556, 257, 158, 157, 66, 58, 55, 848, 778, 693, 680, 604, 412, 278, 250, 134, 842, 824, 816, 733, 718, 676, 648, 519, 438, 374, 356, 353, 312, 163, 67, 749, 742, 731, 725, 722, 653, 633, 609, 345, 226, 54, 998, 828, 505, 101, 947, 705, 670, 536, 418, 269, 813, 488, 445, 409, 201, 155, 59, 47, 905, 771, 677, 664, 469, 446, 381, 240, 193, 899, 885, 876, 818, 787, 782, 767, 730, 461, 73, 902, 868, 810, 618, 499, 326, 189, 784, 700, 600, 493, 416, 248, 215, 702, 658, 550, 282, 231, 681, 587, 359, 389, 837, 750, 744, 638, 516, 60]

**FGVCAircraft**

**Base categories names**: ['Eurofighter Typhoon', 'Hawk T1', 'Spitfire', 'F-16A/B', 'DH-82', 'C-130', 'A380', 'F/A-18', 'Cessna 208', 'Il-76', 'Embraer Legacy 600', 'BAE 146-200', 'ATR-72', 'Global Express', 'DC-3', 'A318', '777-300', 'A310', 'DC-8', 'DHC-1', 'Challenger 600', 'A340-600', 'A340-200', 'Fokker 50', 'Falcon 2000', 'MD-11', 'Gulfstream V', 'A319', 'Fokker 70', 'DC-10', 'A330-300', 'A320', '777-200', 'SR-20', 'DHC-6', 'Cessna 172', 'DHC-8-100', 'DC-6', 'Beechcraft 1900', '707-320', 'Cessna 560', 'A340-300', 'DC-9-30', 'Fokker 100', 'Cessna 525', '747-300', '727-200', 'Metroliner', 'Yak-42', 'Tu-134', 'Saab 340', 'Saab 2000', 'PA-28', 'ERJ 145', 'DHC-8-300', 'C-47', 'ATR-42', 'A330-200', '767-200', 'BAE 146-300', '757-200', 'Model B200', 'MD-90', 'Falcon 900', 'Dornier 328', 'A340-500', '747-400', '747-100', '737-400', 'MD-80']

**Novel categories names**:['Gulfstream IV', 'CRJ-200', 'Boeing 717', '747-200', '737-800', 'Tu-154', 'Tornado', 'MD-87', 'L-1011', 'ERJ 135', 'EMB-120', 'E-195', 'E-190', 'E-170', 'DR-400', 'CRJ-900', 'CRJ-700', 'BAE-125', 'An-12', 'A321', 'A300B4', '767-400', '767-300', '757-300', '737-900', '737-700', '737-600', '737-500', '737-300', '737-200']

**OxfordPets**

**Base categories names**: ['shiba_inu', 'samoyed', 'pug', 'keeshond', 'wheaten_terrier', 'newfoundland', 'german_shorthaired', 'sphynx', 'pomeranian', 'chihuahua', 'saint_bernard', 'russian_blue', 'basset_hound', 'scottish_terrier', 'yorkshire_terrier', 'japanese_chin', 'havanese', 'bengal', 'great_pyrenees', 'beagle', 'miniature_pinscher', 'english_cocker_spaniel', 'siamese', 'leonberger', 'english_setter', 'american_bulldog', 'boxer', 'abyssinian', 'british_shorthair']

**Novel categories names**:['maine_coon', 'egyptian_mau', 'american_pit_bull_terrier', 'staffordshire_bull_terrier', 'ragdoll', 'persian', 'birman', 'bombay']

**SUN397**

**Base categories names**: ['indoor florist_shop', 'skatepark', 'raft', 'oilrig', 'ball_pit', 'martial_arts_gym', 'courtroom', 'cockpit', 'airplane_cabin', 'volcano', 'sauna', 'music_studio', 'indoor volleyball_court', 'batters_box', 'wind_farm', 'wave', 'rock_arch', 'raceway', 'outdoor track', 'oast_house', 'limousine_interior', 'indoor cloister', 'cemetery', 'carrousel', 'baseball stadium', 'auto_factory', 'vineyard', 'toll_plaza', 'television_studio', 'outdoor tennis_court', 'outdoor oil_refinery', 'manufactured_home', 'lift_bridge', 'indoor pilothouse', 'forest_road', 'exterior covered_bridge', 'coral_reef underwater', 'bowling_alley', 'bamboo_forest', 'aquarium', 'veterinarians_office', 'vegetation desert', 'outdoor hangar', 'dining_car', 'control_room', 'barrel_storage wine_cellar', 'squash_court', 'sky', 'promenade_deck', 'playground', 'platform train_station', 'pantry', 'outdoor lido_deck', 'outdoor ice_skating_rink', 'outdoor control_tower', 'kindergarden_classroom', 'kasbah', 'islet', 'indoor brewery', 'igloo', 'heliport', 'courthouse', 'rope_bridge', 'rice_paddy', 'racecourse', 'pulpit', 'landing_deck', 'indoor gymnasium', 'indoor cavern', 'indoor casino', 'ice_floe', 'crevasse', 'butte', 'bus_interior', 'boxing_ring', 'topiary_garden', 'ski_resort', 'pharmacy', 'outdoor greenhouse', 'outdoor athletic_field', 'orchard', 'lighthouse', 'indoor wrestling_ring', 'indoor tennis_court', 'indoor swimming_pool', 'fire_station', 'closet', 'bottle_storage wine_cellar', 'boardwalk', 'outdoor labyrinth', 'landfill', 'indoor jail', 'iceberg', 'bullring', 'art_gallery', 'anechoic_chamber', 'amusement_park', 'videostore', 'throne_room', 'slum', 'sandbox', 'picnic_area', 'outdoor tent', 'laundromat', 'indoor warehouse', 'indoor ice_skating_rink', 'hot_spring', 'exterior gazebo', 'dam', 'campus', 'aqueduct', 'windmill', 'water_tower', 'subway_interior', 'phone_booth', 'pagoda', 'indoor escalator', 'indoor badminton_court', 'establishment poolroom', 'discotheque', 'childs_room', 'archive', 'amphitheater', 'shop bakery', 'riding_arena', 'residential_neighborhood', 'outdoor volleyball_court', 'outdoor general_store', 'outdoor basketball_court', 'interior elevator', 'indoor synagogue', 'indoor firing_range', 'gas_station', 'electrical_substation', 'driveway', 'classroom', 'basilica', 'schoolhouse', 'physics_laboratory', 'outdoor podium', 'mausoleum', 'fountain', 'excavation', 'dorm_room', 'cheese_factory', 'viaduct', 'utility_room', 'outdoor outhouse', 'outdoor driving_range', 'outdoor doorway', 'music_store', 'marsh', 'locker_room', 'kitchenette', 'kitchen', 'indoor shopping_mall', 'indoor booth', 'canyon', 'badlands', 'south_asia temple', 'shoe_shop', 'sandbar', 'sand desert', 'restaurant_kitchen', 'outdoor bazaar', 'indoor market', 'conference_room', 'butchers_shop', 'banquet_hall', 'vegetable_garden', 'railroad_track', 'patio', 'outdoor hot_tub', 'medina', 'hospital_room', 'harbor', 'frontseat car_interior', 'creek', 'chalet', 'campsite', 'boathouse', 'biology_laboratory', 'barn', 'tree_farm', 'snowfield', 'outdoor observatory', 'indoor parking_garage', 'indoor bow_window', 'fishpond', 'elevator_shaft', 'cafeteria', 'broadleaf forest', 'beach', 'train_railway', 'server_room', 'pasture', 'outdoor market', 'indoor hangar', 'golf_course', 'food_court', 'corridor', 'bedroom', 'valley', 'urban canal', 'restau-

rant_patio', 'public atrium', 'outdoor nuclear_power_plant', 'office cubicle', 'indoor pub', 'highway', 'engine_room', 'dining_room', 'crosswalk', 'computer_room', 'tree_house', 'rainforest', 'outdoor bow_window', 'outdoor apartment_building', 'lecture_room', 'indoor stage', 'indoor library', 'indoor jacuzzi', 'indoor chicken_coop', 'indoor bazaar', 'hospital', 'hayfield', 'football stadium', 'beauty_salon', 'skyscraper', 'putting_green', 'operating_room', 'indoor bistro', 'garbage_dump', 'formal_garden', 'dock', 'corn_field', 'construction_site', 'ballroom', 'baggage_claim', 'art_studio', 'wheat_field', 'sushi_bar', 'supermarket', 'ski_lodge', 'runway', 'park', 'outdoor kennel', 'outdoor diner', 'lobby', 'indoor general_store', 'exterior balcony', 'watering_hole', 'van_interior', 'plaza', 'outdoor arrival_gate', 'fire_escape', 'fairway', 'water moat', 'village', 'street', 'shower', 'outdoor planetarium', 'outdoor church', 'jail_cell', 'indoor church', 'indoor cathedral', 'candy_store', 'ticket_booth', 'staircase', 'outdoor power_plant', 'office_building', 'indoor garage', 'catacomb', 'amusement_arcade', 'plunge waterfall', 'jewelry_shop', 'forest_path']

**Novel categories names**:['east_asia temple', 'dentists_office', 'castle', 'bookstore', 'arch', 'alley', 'toyshop', 'pond', 'platform subway_station', 'palace', 'outdoor chicken_coop', 'motel', 'ice_cream_parlor', 'home_office', 'clothing_store', 'auditorium', 'wet_bar', 'tower', 'swamp', 'shopfront', 'parlor', 'outdoor swimming_pool', 'outdoor mosque', 'outdoor cathedral', 'mountain_snowy', 'indoor diner', 'fastfood_restaurant', 'cultivated field', 'parking_lot', 'natural lake', 'herb_garden', 'basement', 'sea_cliff', 'indoor kennel', 'home poolroom', 'game_room', 'fan waterfall', 'conference_center', 'coast', 'bathroom', 'barndoor', 'office', 'indoor factory', 'ice_shelf', 'delicatessen', 'courtyard', 'bridge', 'abbey', 'veranda', 'ski_slope', 'shed', 'indoor mosque', 'indoor greenhouse', 'gift_shop', 'cottage_garden', 'playroom', 'outdoor monastery', 'indoor museum', 'outdoor cabin', 'indoor apse', 'hill', 'burial_chamber', 'berth', 'bar', 'airport_terminal', 'yard', 'stable', 'recreation_room', 'outdoor parking_garage', 'corral', 'thriftshop', 'natural canal', 'indoor movie_theater', 'house', 'attic', 'trench', 'ruin', 'outdoor hunting_lodge', 'interior balcony', 'home dinette', 'building_facade', 'boat_deck', 'river', 'ocean', 'hotel_room', 'baseball_field', 'cliff', 'botanical_garden', 'waiting_room', 'mountain', 'lock_chamber', 'indoor podium', 'door elevator', 'coffee_shop', 'bayou', 'chemistry_lab', 'assembly_line', 'youth_hostel', 'pavilion', 'industrial_area', 'galley', 'art_school', 'reception', 'outdoor hotel', 'living_room', 'wild field', 'outdoor inn', 'outdoor synagogue', 'indoor_procenium theater', 'restaurant', 'nursery', 'needleleaf forest', 'mansion', 'indoor_seats theater', 'drugstore', 'block waterfall', 'vehicle dinette', 'outdoor library', 'clean_room', 'backseat car_interior' ]

## UCF101

**Base categories names**: ['Typing', 'Table_Tennis_Shot', 'Soccer_Penalty', 'Playing_Guitar', 'Military_Parade', 'Ice_Dancing', 'Bowling', 'Blowing_Candles', 'Billiards', 'Bench_Press', 'Field_Hockey_Penalty', 'Baby_Crawling', 'Writing_On_Board', 'Basketball_Dunk', 'Horse_Race', 'Sumo_Wrestling', 'Surfing', 'Clean_And_Jerk', 'Pull_Ups', 'Rock_Climbing_Indoor', 'Playing_Violin', 'Playing_Piano', 'Apply_Eye_Makeup', 'Horse_Riding', 'Sky_Diving', 'Tai_Chi', 'Rafting', 'Playing_Dhol', 'Breast_Stroke', 'Fencing', 'Cutting_In_Kitchen', 'Punch', 'Golf_Swing', 'Playing_Sitar', 'Band_Marching', 'Biking', 'Mopping_Floor', 'Shaving_Beard', 'Uneven_Bars', 'Handstand_Pushups', 'Brushing_Teeth', 'Baseball_Pitch', 'Rowing', 'Blow_Dry_Hair', 'Tennis_Swing', 'Drumming', 'Diving', 'Archery', 'Playing_Flute', 'Walking_With_Dog', 'Skate_Boarding', 'Cliff_Diving', 'Boxing_Punching_Bag', 'Knitting', 'Cricket_Shot', 'Playing_Cello', 'Skiing', 'Playing_Tabla', 'Hula_Hoop', 'Haircut', 'Pommel_Horse', 'Trampoline_Jumping', 'Skijet', 'Basketball', 'Salsa_Spin', 'Long_Jump', 'Apply_Lipstick', 'Volleyball_Spiking', 'Juggling_Balls', 'Floor_Gymnastics']

**Novel categories names**:['High_Jump', 'Front_Crawl', 'Pole_Vault', 'Hammer_Throw', 'Pizza_Tossing', 'Swing', 'Yo_Yo', 'Shotput', 'Head_Massage', 'Jump_Rope', 'Soccer_Juggling', 'Hammering', 'Mixing', 'Kayaking', 'Cricket_Bowling', 'Jumping_Jack', 'Boxing_Speed_Bag', 'Javelin_Throw', 'Handstand_Walking', 'Lunges', 'Push_Ups', 'Throw_Discus', 'Wall_Pushups', 'Nunchucks', 'Frisbee_Catch', 'Body_Weight_Squats', 'Rope_Climbing', 'Parallel_Bars', 'Still_Rings', 'Playing_Daf', 'Balance_Beam']

## Caltech101

**Base categories names**: ['windsor_chair', 'trilobite', 'tick', 'sunflower', 'strawberry', 'stop_sign', 'stegosaurus', 'soccer_ball', 'rooster', 'pyramid', 'pizza', 'panda', 'pagoda', 'okapi', 'motorbike', 'metronome', 'laptop', 'inline_skate', 'headphone', 'gramophone', 'ewer', 'dollar_bill', 'dalmatian', 'car_side', 'cannon', 'buddha', 'brain', 'bonsai', 'barrel', 'accordion', 'airplane', 'watch',

Table 1: Quantitative results on FGVCAircraft of different models under different few shots setting.

| Datasets | Model | Few-shot settings | | | | |
|---|---|---|---|---|---|---|
| | | 1 | 2 | 4 | 8 | 16 |
| FGVCAircraft | Tip-Adapter | 1.30 | 3.29 | 6.18 | 7.98 | 14.86 |
| | Meta-Adapter | 6.88 | 11.67 | 10.77 | 13.26 | 17.05 |
| OxfordPets | Tip-Adapter | 71.14 | 71.56 | 74.12 | 74.47 | 76.69 |
| | Meta-Adapter | 62.08 | 65.58 | 69.60 | 73.70 | 76.86 |
| SUN397 | Tip-Adapter | 31.12 | 34.00 | 37.10 | 42.35 | 44.55 |
| | Meta-Adapter | 35.35 | 40.28 | 43.57 | 48.92 | 48.33 |
| UCF101 | Tip-Adapter | 39.30 | 40.61 | 40.79 | 41.14 | 40.26 |
| | Meta-Adapter | 48.51 | 49.82 | 49.56 | 48.86 | 47.72 |
| Caltech101 | Tip-Adapter | 66.32 | 69.72 | 72.67 | 73.12 | 73.26 |
| | Meta-Adapter | 67.36 | 68.39 | 75.04 | 76.51 | 76.07 |
| DTD | Tip-Adapter | 9.81 | 12.78 | 26.48 | 31.67 | 41.67 |
| | Meta-Adapter | 25.00 | 29.63 | 44.26 | 43.52 | 49.07 |
| EuroSAT | Tip-Adapter | 7.69 | 30.31 | 42.51 | 70.43 | 69.80 |
| | Meta-Adapter | 19.10 | 46.16 | 51.29 | 72.20 | 70.82 |

'starfish', 'helicopter', 'revolver', 'ferry', 'joshua_tree', 'yin_yang', 'wheelchair', 'nautilus', 'emu', 'grand_piano', 'stapler', 'pigeon', 'menorah', 'water_lilly', 'saxophone', 'cougar_face', 'platypus', 'garfield', 'binocular', 'sea_horse', 'cup', 'kangaroo', 'hedgehog', 'bass', 'hawksbill', 'camera', 'umbrella', 'cougar_body', 'dolphin', 'scorpion', 'minaret', 'llama', 'wrench', 'scissors', 'butterfly', 'snoopy', 'euphonium', 'ceiling_fan']

**Novel categories names**: ['beaver', 'leopard', 'mayfly', 'ibis', 'brontosaurus', 'elephant', 'schooner', 'flamingo_head', 'gerenuk', 'flamingo', 'mandolin', 'crocodile', 'chandelier', 'face', 'crayfish', 'anchor', 'rhino', 'lamp', 'lotus', 'dragonfly', 'electric_guitar', 'wild_cat', 'octopus', 'cellphone', 'lobster', 'ketch', 'ant', 'chair', 'crab', 'crocodile_head']

**DTD**

**Base categories names**: ['paisley', 'knitted', 'chequered', 'bubbly', 'crystalline', 'cobwebbed', 'striped', 'pleated', 'cracked', 'studded', 'waffled', 'polka-dotted', 'freckled', 'perforated', 'honeycombed', 'stratified', 'potholed', 'swirly', 'porous', 'grid', 'frilly', 'sprinkled', 'meshed', 'wrinkled', 'spiralled', 'marbled', 'scaly', 'blotchy', 'gauzy', 'woven', 'veined', 'crosshatched']

**Novel categories names**:['braided', 'dotted', 'matted', 'flecked', 'smeared', 'grooved', 'lined', 'banded', 'stained', 'interlaced', 'fibrous', 'zigzagged', 'pitted', 'lacelike', 'bumpy']

**EuroSAT**

**Base categories names**: ['Forest', 'Industrial Buildings', 'Highway or Road', 'Residential Buildings', 'Pasture Land', 'Permanent Crop Land', 'Sea or Lake']

**Novel categories names**:['River', 'Herbaceous Vegetation Land', 'Annual Crop Land']

## 2 More Quantitative Results

In this section, we present the comparison of Meta-Adapter and other methods on the remaining seven datasets under different few-shot settings in Table 1.

## 3 Additional Experiments

We provide the comparison of Meta-Adapter with the SOTA prompt-learning method, CoCoOp [9] in Figure 1. All experiments are conducted under the 16-shot setting. It is clear that Meta-Adapter demonstrates superior generalizability over CoCoOp by large margins.

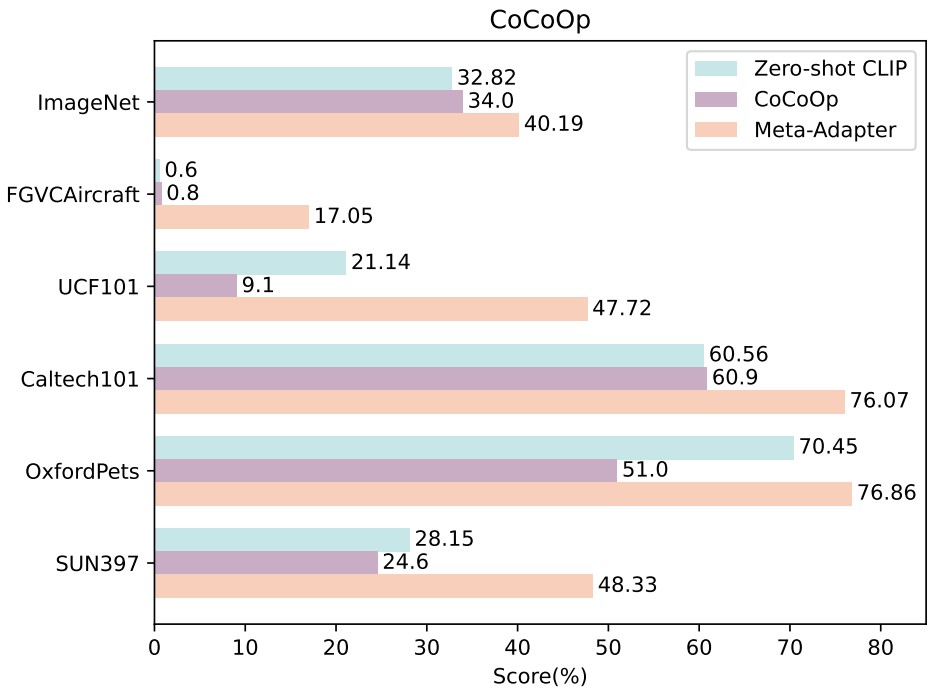

Figure 1: Comparison with CoCoOp.