# OpenReview forum: "Meta-Adapter: An Online Few-shot Learner for Vision-Language Model"
_NeurIPS.cc/2023/Conference — NeurIPS 2023 poster_

### Official Review · Reviewer_Qojd · 2023-07-04

**Soundness:** 3 good
**Presentation:** 3 good
**Contribution:** 3 good
**Rating:** 5
**Confidence:** 4

**Summary:**

For the Vision-Language Model task, which usually requires a small number of samples for fine-tuning, this work proposes a Meta-Adapter method. The Meta-Adapter method is based on the gated multi-head attention mechanism, and can be generalized to unseen categories without additional fine-tuning after a small amount of training by Few-Shot. The effectiveness of this work is experimentally demonstrated.

**Strengths:**

1. From the experimental results, this work performs better in corss-dataset generalization and has good performance on downstream tasks.
2.	This work is well-written, the related works are well summarized and the contributions are clearly demonstrated. The empirical performance is promising.


**Weaknesses:**

1. The explanation of the formulas in this work could be more clear to help the reader understand the content.
2. The connection between the meta-learning method and Meta-Adapter is not well explained.



**Questions:**

1. Why is it called Meta-Adapter? In my understanding, the method of this work is to generalize to new samples by using a small number of training samples, and this method is not called meta-learning.
2. The motivation for this work is that "few-shot learning methods based on CLIP typically require offline fine-tuning of the parameters on few-shot samples ", and the same motivation for Tip-Adapter. It seems to me that Meta-Adapter is an improved approach based on Tip-Adapter, is there a new improvement in the motivation for the approach to this work?

---

> ### Author Rebuttal · Authors · 2023-08-10
>
> **Q1: The explanation of the formulas could be more clear.**
> **A1:**
> Many thanks. Accordingly, we will provide a more comprehensive explanation of the formulas in Section 3.2, including additional descriptions and analyses of the symbols and their functions.
>
> **Q2: Why is it called Meta-Adapter? The connection between the meta-learning method and Meta-Adapter.**
> **A2:**
> The Meta-Adapter aims to build a new clip adapter with general few-shot learning abilities instead of specializing in certain tasks and domains. To achieve it, as stated in Line 46-50, we adopt the meta-testing mechanism. Specifically, we split the data into seen and unseen sets according to category, domain, or task. During training, we randomly sample few-shot image/text pairs from the seen set to optimize the parameters of the meta-adapter, enabling it with general few-shot learning ability. During testing, we first sample some few-shot image/text pairs per category from the unseen set, then fed to the frozen meta-adapter to generate embeddings for each category, and finally predict classification scores by calculating the similarity between the embeddings and features of other images in the unseen set. We will clarify it in the revision.
>
> **Q3: The improvement in the motivation against Tip-Adapter.**
> **A3:**
> The motivation of the Meta-adapter is to learn how to learn from few-shot samples. It has the more general ability to learn in a few-shot setting rather than domain-specific capabilities. Previous methods, including Clip-Adapter and Tip-Adapter, require fine-tuning or hyper-parameter search tailored to the target domain in order to achieve good performance. Our method, on the other hand, can be applied to other domains or tasks seamlessly and demonstrates significant efficacy and performance advantages.

---

> > ### Comment · Reviewer_Qojd · 2023-08-19
> > **Post rebuttal**
> >
> > I am very grateful to the author for his efforts in the rebuttal process. The author's rebuttal partly resolved my doubts, and I decided to maintain the previous rating unchanged.

---

### Official Review · Reviewer_uypu · 2023-07-07

**Soundness:** 2 fair
**Presentation:** 3 good
**Contribution:** 2 fair
**Rating:** 5
**Confidence:** 4

**Summary:**

This paper proposes a meta-adapter structure for CLIP like vision-language backbone. Specifically, a cross-attention with a gate mechanism are used to construct the meta-adapter. It aims to improve the few-show learning ability for current CLIP backbone. Compared with other baselines such as CLIP-adapter and Tip-adapter, the proposed meta-adapter obtains improvements on several experimental settings.

**Strengths:**

1. This paper studies a popular question about CLIP few-shot learning.
2. The motivation and idea are well presented.
3. Meta-adapter is straightforward and easy to follow.

**Weaknesses:**

1. The meta-adapter uses a cross-attention and gate module, which are commonly used for this field resulting in the limited technical contribution. Simply adding these modules to improve the final performance is not very surprising.
2. The proposed module is general for vision-language model, but the evaluation is only for classification. Supplementing retrieval evaluation will further support this method.
3. Some figures are relatively misleading the readers. For example, in figure1, please unify the number scales in the performance axis. Current version is a little confusing to show the performance gain.
4. As shown in Tab1 and the limitation section discussed by the authors, the general performance gain is not very significant.

**Questions:**

Please check the weaknesses.

**Limitations:**

Discussed by the authors.

---

> ### Author Rebuttal · Authors · 2023-08-10
>
> **Q1: Limited technical contribution of the meta-adapter.**
> **A1:**
> The main contribution of this paper is to introduce meta-learning into clip adapters for the first time, to achieve online few-shot learning for visual-language models. We mainly pursue building a new framework for clip adapters, rather than the network architecture. Although the structure of the Meta-Adapter is simple, as shown in Figure 1, it not only has higher efficiency but also stronger generalization performance than the Tip-Adapter. Besides, as shown in Table 7, compared to other sophisticated offline methods, our method demonstrates clear advantages in generalization capability and overall performance.
>
> **Table 7. Comparison of Zero-Shot CLIP, CLIP-Adapter, CoOp, CoCoOp, and Meta-Adapter on ImageNet, UCF101, Caltech101, DTD, and FGVCAircraft datasets in the in-domain generalization setting. H: Harmonic mean.**
>
> | Dataset        | ImageNet | ImageNet | ImageNet | UCF101 | UCF101 | UCF101 | Caltech101 | Caltech101 | Caltech101 | DTD  | DTD   | DTD  | FGVCAircraft | FGVCAircraft | FGVCAircraft |
> |----------------|----------|----------|----------|--------|--------|--------|------------|------------|------------|------|-------|------|--------------|--------------|--------------|
> | Model          | Base     | Novel    | **H**        | Base   | Novel  | **H**      | Base       | Novel      | **H**          | Base | Novel | **H**    | Base         | Novel        | **H**            |
> | Zero-shot CLIP | 71.9     | 32.8     | 45.0     | 79.4   | 21.1   | 33.4   | 95.4       | 60.6       | 74.1       | 59.3 | 8.2   | 14.3 | 23.9         | 0.6          | 1.2          |
> | CLIP-Adapter   | 76.3     | 15.1     | 25.3     | 89.4   | 5.4    | 10.2   | 97.3       | 39.3       | 54.0       | 70.2 | 2.0   | 3.9  | 32.1         | 0.3          | 0.6          |
> | CoOp           | 75.3     | 2.7      | 5.2      | 89.3   | 1.0    | 2.0    | 97.2       | 31.3       | 47.4       | 71.6 | 1.5   | 2.9  | 32.7         | 0.3          | 0.6          |
> | CoCoOp         | 75.5     | 33.9     | 46.8     | 86.5   | 9.1    | 16.5   | 96.8       | 60.9       | 74.8       | 69.1 | 3.0   | 5.8  | 30.0         | 0.8          | 1.6          |
> | Meta-Adapter   | 76.3     | 40.8     | **53.2**     | 82.4   | 47.7   | **60.4**    | 94.9       | 76.1       | **84.4**       | 64.1 | 49.1  | **55.6** | 30.8         | 17.1         | **21.9**         |
>
>
> **Q2: The evaluation is only for classification.**
> **A2:**
> As shown in Table 5 and Section 4.3 in the manuscript, we evaluate our Meta-Adapter for object detection and segmentation tasks. Without further fine-tuning, our method demonstrates superior generalization capabilities, achieving much higher performance than the Tip-Adapter and zero-shot baseline. Additionally, in Table 8 we provide the quantitative results (recall@10) of retrieval evaluation on several classification datasets. The results show that our Meta-Adapter obtains consistent and considerable improvements over Tip-Adapter across datasets.
>
>
> **Table 8. Average Recall of Zero-Shot CLIP and Meta-Adapter on several datasets for retrieval evaluation.**
>
>
>
> |Recall@10       | ImageNet      | FGVC | Oxford Pets |  SUN397 | UCF101 | DTD |
> | -----------    | :-----------: | :-----------: | :-----------: |  :-----------: | :-----------: |  :-----------: |
> |  Zero-shot CLIP |57.5          | 15.1  | 77.0        | 51.9     |  52.4     |   29.8  |
> | Tip-Adapter     |62.5          | 18.8  |77.3        | 57.8     |  59.0     |   39.3  |
> | Meta-Adapter    |**64.2**| **22.4**     | **77.6**          | **61.5**      |  **64.5**    |**46.0**  |
>
>
>
>
> **Q3: Some figures are relatively misleading to the readers.**
> **A3:**
> Thanks for the suggestion. We will unify the number scales of Figure 1 in the revision.
>
> **Q4: The general performance gain is not very significant in Table 1.**
> **A4:**
> Please note that our Meta-Adapter improves over Tip-Adapter by 4.96% on average, while having lower inference cost. Additionally, $\Delta$ represents the gap between optimization on individual datasets and cross-dataset generalization performance. The results reflect that our algorithm has strong generalization capabilities, enabling the cross-dataset performance to be comparable to individually optimized performance. In contrast, Tip-Adapter not only has inferior absolute performance but also notably weaker generalization abilities.

---

> > ### Comment · Reviewer_uypu · 2023-08-18
> > **Post rebuttal**
> >
> > Thank for the authors' feedback. Even if I agree this draft focuses on a new framework instead of structure, I still concern the proposed framework is more like a technical combination. Thank you for reminding the other experimental setting other than classification, actually, my concerns was more about if the retrieval task can be added.
> >
> > I also checked other reviews. My concerns have been partially resolved and I hope the authors can revise/supplement this draft accordingly. I raise my score to 5.
> >
> > Thanks!

---

### Official Review · Reviewer_iC61 · 2023-07-09

**Soundness:** 3 good
**Presentation:** 3 good
**Contribution:** 3 good
**Rating:** 6
**Confidence:** 4

**Summary:**

The main goal of the paper is to explore an approach that is light-weight to allow a CLIP-pretained model to perform well in few-shot settings. The proposed approach (called Meta-Adapter) essentially learns an additional multi-head attention network with an additional gating function. The approach is simple and seems to achieve interesting results.

**Strengths:**

The addressed problem is of great practical relevance and pretraining with CLIP is highly popular these days

Overall the idea is quite simple (which is positive) and intuitive

The experiments give a variety of comparisons to zero-shot CLIP and TIP-Adapter

**Weaknesses:**

The paper mentions on line 61 that the paper performs abalation studies - but honestly the reported experiments in the main paper are not real proper ablations - in particular the paper should have ablated the design choices of the approach where I could not find any experiment - since the approach is so simple that would have been very easy to do and would have helped to understand the approach better

The paper reports slightly lower "base class" performance (e.g. table) 2 for most datasets - which is then "compensated" for by higher "novel class" performance. This is a classic trade-off many works have - but I would have liked to not just see a single pair of base/novel class performance but rather a curve/set of novel/base class performances - that would be again more interesting and telling



**Questions:**

see weaknesses section above

The authors argue that the work is doing some sort of "meta-learning" (lines 116-118) - however, section 3 (method) does not talk about any meta-learning setting as far as I can tell - can you please give an explanation what the meta-learning part is here?

**Limitations:**

ok for me

---

> ### Author Rebuttal · Authors · 2023-08-10
>
> **Q1: More ablation studies of the Meta-Adapter.**
> **A1:**
> Thanks for this insightful suggestion. Accordingly, we further conduct more ablation studies to demonstrate the advantages of our design. As shown in Table 7, the results demonstrate that multi-head attention contributes most significantly to improving accuracy. The proposed learnable gating block can further enhance the performance while introducing value projection leads to decreased generalization capability. Besides, as shown in Table 8, we increase the model scale of the meta-adapter by widening the projection layers (Wider) or cascading multiple modules (Deeper). The results show that increasing the number of modules can improve the parameter size and accuracy slightly, but brings a significant efficiency decrease. We will add these tables to the revision.
>
> **Table 7. Ablation study on different components. The LGB, VP, and MHA indicate the learnable gating block, value projection layer, and multi-head attention block, respectively.**
>
>
> |Method              | ImageNet      | SUN397       | UCF101        |  DTD          |
> |-----------         | :------:      | :---:        | :----:        |:----:         |
> |Meta-Adapter w/  VP | 25.6          |18.2          | 5.9           | 5.0           |
> |Meta-Adapter w/o MHA| 32.9          |28.9          | 21.4          | 10.0          |
> |Meta-Adapter w/o LGB| 40.2          |51.3          | 50.2          | 55.0          |
> |Meta-Adapter        | **40.8**          |**52.6**          |**51.0**          | **55.8**          |
>
>
> **Table 8. Quantitative results of Meta-Adapter's variants on several datasets.**
>
>
> |Method       | ImageNet       | SUN397       | UCF101     |  DTD       |  #Param  | Latency |
> | ----------- | :-----------:  | :-----------:| :---------:|  :-------: | :-------:|  :----: |
> |Meta-Adapter | 40.8           |  52.6        | 51.0       | 55.8       | 2.1M     |  3ms    |
> |Wider (X2)   | 40.1           |  51.4        | 50.0       | 55.0       | 4.2M     |  5ms    |
> |Wider (X4)   | 40.5           |  51.3        | 50.4       | 55.0       | 8.4M     |  9ms    |
> |Deeper (X2)  | 40.4           |  52.9        | 52.2       | 56.7       | 4.2M     |  6ms    |
> |Deeper (X4)  | 38.9           |  52.7        | 51.4       | 56.3       | 8.4M     |  11ms   |
>
>
>
> **Q2: The performance trade-off between base and novel classes.**
> **A2:**
> Thanks for this valuable suggestion. To demonstrate the generalization ability of Meta-Adapter, similar to CoCoOp [22], we introduce the harmonic mean as the evaluation criteria, which is used to reflect the overall performance on both seen and unseen sets [57]. As shown in Table 9, although Tip-Adapter can overfit the seen data on some datasets, its overall performance is significantly inferior to Meta-Adapter. We will update the table in the revision.
>
> **Table 9. Comparison of Zero-Shot CLIP, Tip-Adapter, and Meta-Adapter on ImageNet, UCF101, Caltech101, DTD, and FGVCAircraft datasets in the in-domain generalization setting. H: Harmonic mean.**
>
> | Dataset        | ImageNet | ImageNet | ImageNet | UCF101 | UCF101 | UCF101 | Caltech101 | Caltech101 | Caltech101 | DTD  | DTD   | DTD  | FGVCAircraft | FGVCAircraft | FGVCAircraft |
> |----------------|----------|----------|----------|--------|--------|--------|------------|------------|------------|------|-------|------|--------------|--------------|--------------|
> | Model          | Base     | Novel    | **H**        | Base   | Novel  | **H**      | Base       | Novel      | **H**          | Base | Novel | **H**    | Base         | Novel        | **H**            |
> | Zero-shot CLIP | 71.9     | 3.8     | 45.0     | 79.4   | 21.1   | 33.4   | 95.4       | 60.6       | 74.1       | 59.3 | 8.2   | 14.3 | 23.9         | 0.6          | 1.2          |
> |Tip-Adapter  |       73.3         | 36.5          | 48.7     | 85.2     | 40.3      | 54.7      | 96.3   | 73.3    | 83.2    | 66.8       | 41.7       | 51.3       | 30.7 | 14.9   | 20.0  |
> | Meta-Adapter   | 76.3     | 40.8     | **53.2**     | 82.4   | 47.7   | **60.4**    | 94.9       | 76.1       | **84.4**       | 64.1 | 49.1  | **55.6** | 30.8         | 17.1         | **21.9**         |
>
> [22] Zhou K, Yang J, Loy C C, et al. Conditional prompt learning for vision-language models[C]//Proceedings of the IEEE/CVF Conference on Computer Vision and Pattern Recognition. 2022: 16816-16825.
>
> [57] Xian, Y., Schiele, B. and Akata, Z., 2017. Zero-shot learning-the good, the bad and the ugly. In
> Proceedings of the IEEE conference on computer vision and pattern recognition (pp. 4582-4591).
>
>
>
> **Q3: Which part explains the meta-learning settings?**
> **A3:**
> Sorry for the confusion. We have presented the meta-learning strategy of our method in Line 46-50. Specifically, we split the data into seen and unseen sets according to category, domain, or task. During training, we randomly sample few-shot image/text pairs from the seen set to optimize the parameters of the meta-adapter, enabling it with general few-shot learning ability. During testing, we first sample some few-shot image/text pairs per category from the unseen set, then fed to the frozen meta-adapter to generate embeddings for each category, and finally predict classification scores by calculating the similarity between the embeddings and features of other images in the unseen set. We will clarify it in the revision.

---

> > ### Comment · Reviewer_iC61 · 2023-08-18
> >
> > Thanks for the response that essentially addresses my main questions. Please include that information also in the final paper !
> >
> > My assessment - also after reading the other reviews and rebuttals - remains the same.

---

### Official Review · Reviewer_JEGz · 2023-07-16

**Soundness:** 2 fair
**Presentation:** 3 good
**Contribution:** 2 fair
**Rating:** 5
**Confidence:** 3

**Summary:**

This paper proposes Meta-Adapter which can refine the CLIP features guided by the few-shot samples in an online manner. The major challenge of adapting CLIP with few-shot samples is over-fitting. Compared with offline approaches CoOp or online approaches TIP-Adapter, Meta-Adapter alleviates the over-fitting problem and demonstrates superior generalization across datasets. The author further adapts the Meta-Adapter to an open-vocabulary object detector, ViLD, and also finds decent improvements

**Strengths:**

+ This paper is generally well-written and easy to follow.
+ The proposed Meta-Adapter shows improved generality over different datasets compared with the Tip-adapter baseline.
+ Meta-Adapter can be served as a plugin module for open-vocabulary models, such as ViLD.

**Weaknesses:**

- Limited comparisons with existing approaches. Even though there are few online approaches like Tip-adapter, it is also encouraged to include more competitors, such as offline approaches in the experimental discussions.
- The authors claim that Meta-Adapter uses 'a lightweight residual style adapter'. However, what is the optimal size of the adapter is not justified in ablation.

**Questions:**

- The captions of each figure are short. I recommend adding more detailed explanations for each figure.

---

> ### Author Rebuttal · Authors · 2023-08-10
>
> **Q1: Limited comparisons with existing approaches.**
> **A1:**
> Thanks for the suggestions. As shown in Table 7, we provide the ablation studies between our meta-adapter and other offline methods. For a fair comparison, similar to CoCoOp, the experiments adopt a base-to-novel generalization setting. The results demonstrate that our method has significant advantages in generalization on novel classes, e.g. improving over CoCoOp by 6.9% on ImageNet. Additionally, as in previous works, we introduce Harmonic Mean to measure overall performance on base and novel classes. It can be observed that our approach also shows clear superiority in terms of overall performance. Moreover, as shown in Table 8, we provide training time comparisons between different methods, where the duration is based on default settings in the original papers. The results show that our method not only surpasses these offline approaches in generalization performance but also has noticeable advantages in training speed. More importantly, finetuning is not needed for our method when applying to new datasets or tasks. We will add these tables to the revision.
>
> **Table 7. Comparison of Zero-Shot CLIP, CLIP-Adapter, CoOp, CoCoOp, and Meta-Adapter on ImageNet, UCF101, Caltech101, DTD, and FGVCAircraft datasets in the in-domain generalization setting. H: Harmonic mean.**
>
> | Dataset        | ImageNet | ImageNet | ImageNet | UCF101 | UCF101 | UCF101 | Caltech101 | Caltech101 | Caltech101 | DTD  | DTD   | DTD  | FGVCAircraft | FGVCAircraft | FGVCAircraft |
> |----------------|----------|----------|----------|--------|--------|--------|------------|------------|------------|------|-------|------|--------------|--------------|--------------|
> | Model          | Base     | Novel    | **H**        | Base   | Novel  | **H**      | Base       | Novel      | **H**          | Base | Novel | **H**    | Base         | Novel        | **H**            |
> | Zero-shot CLIP | 71.9     | 32.8     | 45.0     | 79.4   | 21.1   | 33.4   | 95.4       | 60.6       | 74.1       | 59.3 | 8.2   | 14.3 | 23.9         | 0.6          | 1.2          |
> | CLIP-Adapter   | 76.3     | 15.1     | 25.3     | 89.4   | 5.4    | 10.2   | 97.3       | 39.3       | 54.0       | 70.2 | 2.0   | 3.9  | 32.1         | 0.3          | 0.6          |
> | CoOp           | 75.3     | 2.7      | 5.2      | 89.3   | 1.0    | 2.0    | 97.2       | 31.3       | 47.4       | 71.6 | 1.5   | 2.9  | 32.7         | 0.3          | 0.6          |
> | CoCoOp         | 75.5     | 33.9     | 46.8     | 86.5   | 9.1    | 16.5   | 96.8       | 60.9       | 74.8       | 69.1 | 3.0   | 5.8  | 30.0         | 0.8          | 1.6          |
> | Meta-Adapter   | 76.3     | 40.8     | **53.2**     | 82.4   | 47.7   | **60.4**    | 94.9       | 76.1       | **84.4**       | 64.1 | 49.1  | **55.6** | 30.8         | 17.1         | **21.9**         |
>
>
> **Table 8. Training time on ImageNet of Meta-Adapter and other offline methods on a single GeForce RTX 3090.**
>
>
>
> |CLIP-Adapter (200 epochs)        | CoOp (200 epochs)      | CoCoOp (10 epochs) | Meta-Adapter (10 epochs)  |
> | -----------    | :-----------: | :-----------: | :-----------:  |
> |  17h 30min | 15h 10min          |  23h 30min         | **20min**           |
>
>
> **Q2: The captions of each figure are short.**
> **A2:**
> Many Thanks. We will add more details to the captions according to the reviewer's suggestion.
>
> **Q3: What is the optimal size of the adapter is not justified in ablation.**
> **A3:**
> We appreciate the comments. Since we pursue online few-shot learning, our method needs to balance both accuracy and efficiency. As shown in Table 9, we increase the model scale of the meta-adapter by widening the projection layers (Wider) or cascading multiple modules (Deeper). The results show that increasing the number of modules can improve the parameter size and accuracy slightly, but brings a significant efficiency decrease. We will clarify it and add this table to the revision.
>
> **Table 9. Quantitative results of Meta-Adapter's variants on several datasets.**
>
>
> |Method       | ImageNet       | SUN397       | UCF101     |  DTD       |  #Param  | Latency |
> | ----------- | :-----------:  | :-----------:| :---------:|  :-------: | :-------:|  :----: |
> |Meta-Adapter | 40.8           |  52.6        | 51.0       | 55.8       | 2.1M     |  3ms    |
> |Wider (X2)   | 40.1           |  51.4        | 50.0       | 55.0       | 4.2M     |  5ms    |
> |Wider (X4)   | 40.5           |  51.3        | 50.4       | 55.0       | 8.4M     |  9ms    |
> |Deeper (X2)  | 40.4           |  52.9        | 52.2       | 56.7       | 4.2M     |  6ms    |
> |Deeper (X4)  | 38.9           |  52.7        | 51.4       | 56.3       | 8.4M     |  11ms   |

---

> > ### Comment · Reviewer_JEGz · 2023-08-18
> > **Good rebuttal, please keep additional experiments in the paper**
> >
> > Thank the authors for the detailed rebuttal. It resolved all my concerns and I highly encourage the authors to include the additional experiments in the main paper or appendix.
> >
> > I have raised my score to 5: Borderline accept.

---

### Official Review · Reviewer_deHD · 2023-07-26

**Soundness:** 3 good
**Presentation:** 2 fair
**Contribution:** 3 good
**Rating:** 6
**Confidence:** 3

**Summary:**

This paper proposes an online adaptation method for CLIP (no fine-tuning of few-shot samples of unseen categories are required, unlike CoOp, CoCoOp, and CLIP-Adapter), called Meta-Adapter. The main claim seems to be that Meta-Adapter is more robust than the most related approach Tip-Adapter, which relies heavily on the hyperparameter search strategy on the target dataset – alpha and beta in Eq (2) which help adjust the weight between category embeddings and few-shot visual embeddings.

More specially, the high-level idea of Meta-Adapter can be found in Figure 2 and Eq. (3). One can make a comparison between Eq. (2) and Eq. (3) to see how Meta-Adapter simplifies the online adaptation. The architecture of Meta-Adapter is based on the gated multi-head attention mechanism [26].

**Strengths:**

S1: Even though the design choice for Meta-Adapter seems arbitrary, it is simple and sound.

S2: Experimental results show the superiority of Meta-Adapter to Tip-Adapter across tasks/settings and architectures (Table 1-5).

**Weaknesses:**

W1: Experimental settings seem to deviate from those in Tip-Adapter, making them less convincing. Examples: (i) This paper considers 8 image classification datasets instead of 11. (ii) The efficiency analysis is lacking. (iii) Related to (ii), there are no CoOp, CoCoOp, or CLIP-Adapter baselines. (I understand that these are online approaches but it would be nice to discuss, for example, an effectiveness/efficiency tradeoff).

W2: Clarity. This point is minor but overall the paper seems to jump into details too quickly and it would benefit from more coarse-to-fine writing. For example, Section 3.2 could benefit form describing different components before jumping into details.

**Questions:**

N/A

---

> ### Author Rebuttal · Authors · 2023-08-10
>
> **Q1: Experiments on more image classification datasets.**
> **A1:**
> Thanks for the valuable suggestion. As shown in Table 7, we conduct the experiments in the other 3 datasets as the reviewer mentioned. Similar to the reported results in the main paper, our method also achieves consistent gains over the Tip-Adapter. We will add this table to the revision.
>
>
> **Table 7. Quantitative results of Meta-Adapter and other methods on Food101, Stanford Cars, and Oxford Flowers datasets.**
>
>
> |Method          | Food101       | Stanford Cars | Oxford Flowers |
> | -----------    | :-----------: | :-----------: | :-----------:  |
> |  Zero-Shot CLIP| 77.4          |  55.7         | 66.0           |
> |  Tip-Adapter   | 77.8          | 66.7          | 89.9           |
> |  Meta-Adapter  | **79.0** | **67.3**| **93.5** |
>
>
> **Q2: Efficiency analysis is lacking.**
> **A2:**
> We have reported the comparison of inference time for online methods in Figure 1(b). The results show that the inference time of Tip-Adapter increases linearly with the number of input shots, due to its individual encoding for each shot. In contrast, our method can maintain a much lower and constant time consumption, while achieving a higher accuracy.
>
> **Q3: Comparison with offline methods.**
> **A3:**
> As shown in Table 8, we provide the ablation studies between our meta-adapter and other offline methods. For a fair comparison, similar to CoCoOp, the experiments adopt a base-to-novel generalization setting. The results demonstrate that our method has significant advantages in generalization on novel classes, e.g. improving over CoCoOp by 6.9% on ImageNet. Additionally, as in previous works, we introduce Harmonic Mean to measure overall performance on base and novel classes. It can be observed that our approach also shows clear superiority in terms of overall performance. Moreover, as shown in Table 9, we provide training time comparisons between different methods, where the duration is based on default settings in the original papers. The results show that our method not only surpasses these offline approaches in generalization performance but also has noticeable advantages in training speed. More importantly, finetuning is not needed for our method when applying to new datasets or tasks. We will add these tables to the revision.
>
> **Table 8. Comparison of Zero-Shot CLIP, CLIP-Adapter, CoOp, CoCoOp, and Meta-Adapter on ImageNet, UCF101, Caltech101, DTD, and FGVCAircraft datasets in the in-domain generalization setting. H: Harmonic mean.**
>
> | Dataset        | ImageNet | ImageNet | ImageNet | UCF101 | UCF101 | UCF101 | Caltech101 | Caltech101 | Caltech101 | DTD  | DTD   | DTD  | FGVCAircraft | FGVCAircraft | FGVCAircraft |
> |----------------|----------|----------|----------|--------|--------|--------|------------|------------|------------|------|-------|------|--------------|--------------|--------------|
> | Model          | Base     | Novel    | **H**        | Base   | Novel  | **H**      | Base       | Novel      | **H**          | Base | Novel | **H**    | Base         | Novel        | **H**            |
> | Zero-shot CLIP | 71.9     | 32.8     | 45.0     | 79.4   | 21.1   | 33.4   | 95.4       | 60.6       | 74.1       | 59.3 | 8.2   | 14.3 | 23.9         | 0.6          | 1.2          |
> | CLIP-Adapter   | 76.3     | 15.1     | 25.3     | 89.4   | 5.4    | 10.2   | 97.3       | 39.3       | 54.0       | 70.2 | 2.0   | 3.9  | 32.1         | 0.3          | 0.6          |
> | CoOp           | 75.3     | 2.7      | 5.2      | 89.3   | 1.0    | 2.0    | 97.2       | 31.3       | 47.4       | 71.6 | 1.5   | 2.9  | 32.7         | 0.3          | 0.6          |
> | CoCoOp         | 75.5     | 33.9     | 46.8     | 86.5   | 9.1    | 16.5   | 96.8       | 60.9       | 74.8       | 69.1 | 3.0   | 5.8  | 30.0         | 0.8          | 1.6          |
> | Meta-Adapter   | 76.3     | 40.8     | **53.2**     | 82.4   | 47.7   | **60.4**    | 94.9       | 76.1       | **84.4**       | 64.1 | 49.1  | **55.6** | 30.8         | 17.1         | **21.9**         |
>
> **Table 9. Training time on ImageNet of Meta-Adapter and other offline methods on a single GeForce RTX 3090.**
>
> |CLIP-Adapter (200 epochs)        | CoOp (200 epochs)      | CoCoOp (10 epochs) | Meta-Adapter (10 epochs)  |
> | -----------    | :-----------: | :-----------: | :-----------:  |
> |  17h 30min | 15h 10min          |  23h 30min         | **20min**           |
>
> **Q4: Minor issues about clarity.**
> **A4:**
> Thanks for the constructive comments. Accordingly, we will improve the presentation by adding more high-level descriptions and backgrounds before Section 3.2.

---

### Decision · Program_Chairs · 2023-09-21

**Decision:**

Accept (poster)

**Comment:**

This paper receives three borderline accepts and two weak accepts. Initially, the reviewers have common concerns for writing clarity, insufficient experiments (e.g., efficiency analysis, the adapter size ablation, limited comparisons with existing approaches), the connection with meta-learning, etc. The rebuttal answers the questions from the reviewers, and the reviewers' concerns are resolved after the rebuttal. AC gives the recommendation of accept (poster) based on the authors' and reviewers' response, and encourages the authors to include these additional results and improve the paper writing in the revised version.